# Construction of a complete set of *Neisseria meningitidis* mutants and its use for the phenotypic profiling of this human pathogen

Alastair Muir[1], Ishwori Gurung[1], Ana Cehovin[1], Adelme Bazin [2], David Vallenet [2] & Vladimir Pelicic [1✉]

The bacterium *Neisseria meningitidis* causes life-threatening meningitis and sepsis. Here, we construct a complete collection of defined mutants in protein-coding genes of this organism, identifying all genes that are essential under laboratory conditions. The collection, named NeMeSys 2.0, consists of individual mutants in 1584 non-essential genes. We identify 391 essential genes, which are associated with basic functions such as expression and pre-servation of genome information, cell membrane structure and function, and metabolism. We use this collection to shed light on the functions of diverse genes, including a gene encoding a member of a previously unrecognised class of histidinol-phosphatases; a set of 20 genes required for type IV pili function; and several conditionally essential genes encoding anti-toxins and/or immunity proteins. We expect that NeMeSys 2.0 will facilitate the phenotypic profiling of a major human bacterial pathogen.

[1] MRC Centre for Molecular Bacteriology and Infection, Imperial College London, London, UK. [2] LABGeM, Génomique Métabolique, CEA, Genoscope, Institut François Jacob, Université d'Evry, Université Paris-Saclay, CNRS, Evry, France. ✉email: v.pelicic@imperial.ac.uk

Low-cost massively parallel sequencing methods have transformed biology, with the availability of complete genome sequences increasing exponentially. In bacteria, genome sequences can be determined for hundreds of isolates in parallel in just a matter of days. For bacterial pathogens, genome sequences are often available for thousands of clinical isolates and represent a key resource for epidemiology and comparative genomics[1]. This explosion has also led to the identification of millions of new genes, with the important insight that many are of unknown function. Even in a model species such as *Escherichia coli* K-12, perhaps the most thoroughly studied biological entity, only 60% of 4623 protein-coding genes have experimentally verified functions[2], while 35% have either no predicted function or a predicted function that is yet to be verified[2]. In less well-studied species, this partition is dramatically skewed towards the latter class, with an overwhelming majority of genes having predicted but not experimentally verified function or no predicted function at all. This indicates that there are still major gaps in our understanding of basic bacterial physiology, including well-studied metabolic pathways[3]. This has fundamental consequences for systems and synthetic biology, by hindering holistic understanding of the life-supporting functions and processes allowing bacteria to grow and replicate. It also has practical consequences by slowing down the identification of new means for controlling bacterial pathogens, which are needed more than ever in this era of antimicrobial resistance. Therefore, developing methods for systematically elucidating gene function is a critical biological endeavour.

Molecular genetics, which involves the creation and phenotypic analysis of large collections of mutants, is the most direct path to determining gene function on a genome scale by providing a crucial link between phenotype and genotype. In many bacteria, genome-wide mutant collections in protein-coding genes have been constructed using either targeted mutagenesis or random transposon (Tn) mutagenesis. Because of its time and cost-effectiveness, Tn mutagenesis is more widely used[4–6], especially since it has been coupled with massively parallel sequencing of Tn insertion sites (Tn-Seq)[7], which allows simultaneous monitoring of the fitness in pools of hundreds of thousands of mutants[8,9]. However, Tn-Seq is not without drawbacks. (1) Not all intragenic Tn insertions lead to a loss of function. (2) Some properties cannot be assessed in a pooled assay format. (3) Specific mutants cannot be recovered from pools for further individual study. (4) Libraries of Tn mutant are never complete, no matter how large. In contrast, complete collections of mutants constructed by targeted mutagenesis do not suffer from these limitations, but their construction is laborious and expensive. For this reason, such collections are available for only a handful of bacterial species: the model Gram-negative *E. coli* K-12 (Gammaproteobacteria)[10], the model Gram-positive *Bacillus subtilis* (Firmicute)[11], *Acinetobacter baylyi* (Gammaproteobacteria) studied for its metabolic properties and possible biotechnological applications[12], and *Streptococcus sanguinis* (Firmicute) an opportunistic pathogen causing infective endocarditis[13]. The utility of such resources is best exemplified by a large-scale study using the Keio collection of ~4000 mutants in *E. coli* K-12, which identified mutants presenting altered colony size upon growth in more than 300 different conditions[14]. More than 10,000 phenotypes were identified for ~2000 mutants, including many in genes of unknown function[14]. Another major advantage of mutant collections constructed by targeted mutagenesis is that they also reveal complete lists of essential genes, those that cannot be mutated because their loss is lethal. These genes that encode proteins essential for cellular life, many of which are expected to be common to all living organisms[15], have attracted particular interest because they are expected to shed light on the origin of life. Additionally, essential genes specific to bacteria are also interesting because they are prime targets for the design of new drugs to tackle antimicrobial resistance.

Since the few species in which complete collections of mutants have been constructed represent only a tiny fraction of bacterial diversity, additional complete collections of mutants in diverse species are needed to tackle more effectively the challenge of genes of unknown function. Human bacterial pathogens are particularly attractive candidates for this venture because of the potential practical implications for human health. Among these, *Neisseria meningitidis*—a Gram-negative Betaproteobacteria causing life-threatening meningitis and sepsis—stands out as an ideal candidate for several reasons. (1) It has a small genome with ~2000 protein-coding genes, and yet displays a robust metabolism allowing rapid growth (doubling time of 40 min), even on minimal medium. (2) It has been the subject of intensive investigation for decades and there are several thousand publicly available meningococcal genome sequences. (3) It is naturally competent, which makes it a model for molecular genetics. Previously, we have used these advantages to design a modular toolbox for *Neisseria meningitidis* systematic analysis of gene function, named NeMeSys[16]. The central module of NeMeSys was an ordered library of ~4500 Tn mutants in which the Tn insertion sites were sequence-defined[16,17], revealing insertions in more than 900 genes[16].

In the present study, we have (1) used the original library of Tn mutants as a starting point to generate a complete collection of defined mutants in *N. meningitidis*—named NeMeSys 2.0—comprising one mutant in each non-essential protein-coding gene, (2) defined and analysed the essential meningococcal genome and (3) illustrated the potential of NeMeSys 2.0 for the phenotypic profiling of the meningococcal genome by identifying the function of multiple genes of unknown function.

## Results and discussion

**Construction of the complete NeMeSys 2.0 collection of meningococcal mutants.** The present study gave new impetus to a systematic re-annotation of the genome of *N. meningitidis* 8013 published in 2009[16]. We took into account new information in the public databases, new publications, an RNA-Seq analysis performed in 8013[18] and findings in the present study. In brief, we dropped one previously annotated gene, updated the gene product annotations of 521 genes, added 69 non-coding RNAs and changed 176 gene start sites. The genome of *N. meningitidis* 8013 contains 2060 protein-coding genes (labelled NMV_). In parallel, we re-sequenced the genome of the 8013 wild-type (WT) strain by Illumina whole-genome sequencing (WGS), which confirmed the accuracy of the original genome sequence determined by Sanger sequencing more than 11 years ago[16], with an error rate of ~1 per 134,000 bases. We identified 17 differences with the published sequence, which are likely to be genuine since they were also found in the few mutants that were verified by WGS (see below). Of these differences, 11 correspond to single-nucleotide polymorphisms, five are single-nucleotide indels and one is the deletion of a GT dinucleotide within a repeated GT tract in NMV_0677, which therefore corresponds to a phase variation event[19].

In keeping with previous studies describing the construction of complete collections of mutants[10–13], 85 genes (4.1%) were not considered to be meaningful targets for mutagenesis (Fig. 1) because they encode transposases for repeated insertion sequences, or they correspond to short remnants of truncated genes and/or non-expressed cassettes (Supplementary Data 1). We, therefore, set out to assemble a complete collection of mutants in the remaining 1975 genes (95.9%) following a two-step

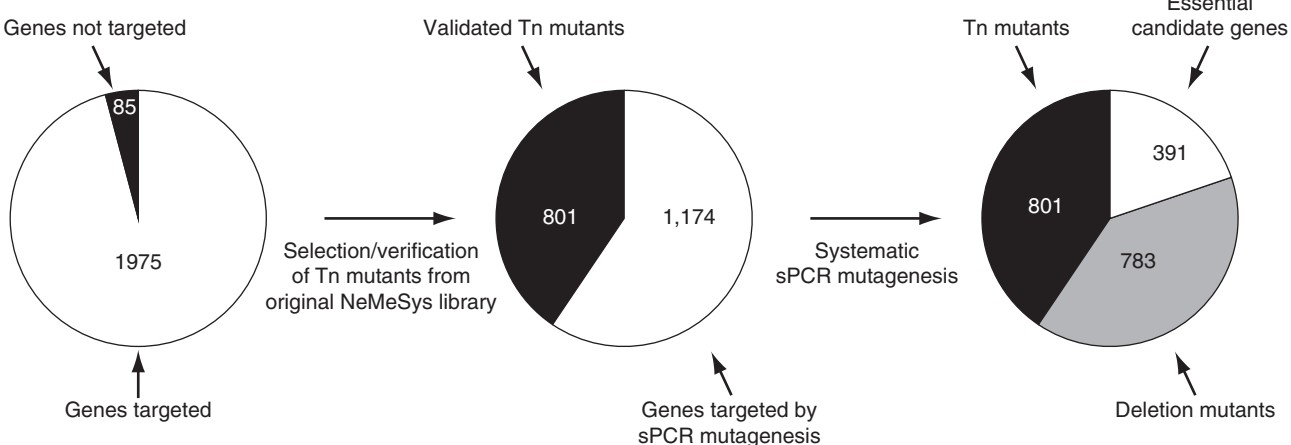

**Fig. 1 Flowchart of the construction of the NeMeSys 2.0 complete collection of mutants in _N. meningitidis_ 8013.** As for similar efforts in other bacteria[10,12,13], we first selected protein-coding genes to be targeted by systematic mutagenesis, excluding 85 genes (4.1%, highlighted in black in the first pie chart) because they encode transposases of repeated insertion sequences, or correspond to short remnants of truncated genes or cassettes (Supplementary Data 1). We then followed a two-step mutagenesis approach explained in the text and in Supplementary Fig. 1. In brief, we first selected a subset of sequence-verified Tn mutants from a previously constructed arrayed library[16,17]. Mutations were re-transformed in strain 8013 and PCR-verified. We thus selected 801 Tn mutants with a disrupting transposon in the corresponding target genes (highlighted in black in the second and third pie charts). Next, we systematically mutagenised the remaining 1174 target genes using a validated no-cloning mutagenesis method relying on sPCR[20]. For each successful transformation, two colonies were isolated and PCR-verified. To minimise false-positive identification of essential genes, each transformation that yielded no transformants was repeated at least three times. In total, we could construct an additional 783 mutants (highlighted in grey in the third pie chart), generating an ordered library of defined mutants in 1584 meningococcal genes (Supplementary Data 3). This effort also identified 391 candidate essential genes, which could not be disrupted, encoding proteins required for _N. meningitidis_ growth on rich medium (Supplementary Data 4).

approach described below. In brief, we first selected a subset of suitable Tn mutants from the NeMeSys library, and then systematically mutagenised the remaining target genes by allelic exchange (Supplementary Fig. 1).

First, since we previously constructed an arrayed library of Tn mutants in strain 8013 in which Tn insertion sites were sequence-defined[16,17], we selected a subset of suitable mutants from this library. Specifically, mutations with a Tn insertion closer to the centre of the genes (excluding the first and last 15% that are often not gene-inactivating) (Supplementary Fig. 1) were first re-transformed into strain 8013, which is naturally competent. After extraction of the genomic DNA of the transformants, we PCR-verified that they contained a Tn inserted in the expected target gene using suitable pairs of flanking primers (Supplementary Data 2). We thus selected a subset of 801 Tn mutants (40.6% of target genes), harbouring Tn insertions expected to be gene-inactivating (Fig. 1) (Supplementary Data 3). As a control, Illumina WGS of one mutant, in _nlaB_, confirmed that it contained the expected mutation, a Tn inserted after a TA dinucleotide at position 2,157,750 in the genome.

Next, we set out to systematically mutagenise the remaining 1174 target genes (59.4%) by allelic exchange. For this, we used a previously validated no-cloning mutagenesis method[20], based on splicing PCR (sPCR), to construct non-polar mutants in which a central portion of the target gene (between the first and last 30%) would be deleted and replaced by the same kanamycin (Km) resistance cassette present in the above Tn (Supplementary Fig. 1). In brief, three PCR products were first amplified, corresponding to the Km cassette and regions upstream and downstream target genes. For failed reactions, we undertook as many rounds of primer design as needed, until all PCRs were successful. The three PCR products were then combined and spliced together. The final sPCR product was transformed directly into _N. meningitidis_ 8013. Typically, for non-essential genes, we obtained hundreds of colonies resistant to kanamycin (Km^R). For

each successful transformation, two Km^R colonies were isolated and PCR-verified to contain the expected mutation. To minimise false-positive identification of essential genes, each transformation that yielded no transformants was repeated at least three times with different sPCR products. Only when all transformations failed to yield transformants, was the target gene deemed to be essential. In summary (Fig. 1), out of the 1174 genes that were targeted, we could disrupt 783 (Supplementary Data 3), while 391 could not be mutated and are thus essential for growth on rich medium (Supplementary Data 4). As above, WGS of a few mutants constructed by sPCR, in _lnt_, _tatB_ and _secB_, confirmed that each contained the expected mutation.

Taken together, using the above two-step approach, we constructed a comprehensive set of 1584 defined mutants in strain 8013, named the NeMeSys 2.0 collection. In addition, we also identified 391 candidate essential genes (19% of all protein-coding genes), which are required for growth of the meningococcus on rich medium. For each gene that was successfully mutated, one mutant was stored at −80 °C in glycerol, while the corresponding genomic DNA was stored at −20 °C. This allows for easy distribution of mutants to the community and/or re-transformation of the corresponding mutations in 8013, other meningococcal strains, or even genetically closely related species like the gonococcus, which share most of their genome with the meningococcus.

**Analysis of the meningococcal essential genome.** Considering their fundamental and practical importance, we first focused on the 391 genes essential for meningococcal life. Since corresponding proteins are expected to be broadly conserved, we determined the partition of the 391 essential genes between conserved and variable genomes in meningococci. To do this, we first used the recently described PPanGGOLiN software[21]—an expectation-maximisation algorithm based on multivariate Bernoulli mixture model coupled with a Markov random field—to

compute the pangenome of *N. meningitidis*, based on complete genome sequences of 108 meningococcal isolates available in RefSeq (Supplementary Data 5). We thereby classified genes in three categories[21], persistent (gene families present in almost all genomes), shell (present at intermediate frequencies), or cloud (present at low frequency). We then determined how the 2060 protein-coding genes of *N. meningitidis* 8013 partitioned between these three classes (Supplementary Data 5). Only 1664 genes (80.8%) belong to the persistent genome, while 396 genes (19.2%) correspond to gene families present at intermediate/low frequency in meningococci since they are part of the shell (241 genes) and cloud genomes (155 genes) (Fig. 2a). Critically, besides highlighting the well-known genomic plasticity of this naturally competent species, this analysis confirmed our original prediction by revealing that essential genes are overwhelmingly conserved in meningococcal isolates. Indeed, 382 essential genes (97.4%) are part of the persistent meningococcal genome (Fig. 2b and Supplementary Data 5). Of the remaining nine essential genes, six are part of the shell genome, while three belong to the cloud genome (Supplementary Data 5).

Furthermore, many meningococcal essential genes (30.2%) are conserved and essential in other bacteria where systematic mutagenesis efforts have been performed (Fig. 3a), including in phylogenetically distant species such as *S. sanguinis* (Supplementary Data 6)[10,12,13]. When conservation was assessed with Proteobacteria only (*E. coli* and *A. baylyi*), which are more closely related to *N. meningitidis*, half of the 391 essential meningococcal genes (195 in total) are conserved and essential in these three Gram-negative species (Fig. 3a). Interestingly, a similar proportion of meningococcal essential genes (175 in total) was conserved in JCVI-syn3.0 (Fig. 3b) (Supplementary Data 7), a synthetic *Mycoplasma mycoides* bacterium engineered with a minimal genome[22] smaller than that of any naturally occurring autonomously replicating cell. Taken together, these observations are consistent with the notion that a sizeable portion of the genome essential for cellular life is widely conserved in bacteria.

Next, to have a more global understanding of the biological functions that are essential in the meningococcus, we analysed the distribution of the 391 essential genes in functional categories (Supplementary Data 8). Genes were classified in specific functional categories using the bioinformatic tools embedded in the MicroScope platform[23], which hosts NeMeSys 2.0. In particular, we used predictions from MultiFun[24], MetaCyc[25], eggNOG[26], COG[27], FIGfam[28] and/or InterProScan[29]. As can be seen in Table 1, the essential meningococcal genes could be distributed in a surprisingly limited number of pathways. In keeping with expectations, many essential genes are involved in key cellular processes such as (1) transcription (11 genes encoding subunits of the RNA polymerase, a series of factors modulating transcription, and two transcriptional regulators), (2) RNA modification/degradation (19 genes) and (3) protein biosynthesis (120 genes) (Table 1). The large class of genes involved in protein biosynthesis encode (1) all 24 proteins involved in tRNA charging, (2) virtually all ribosomal proteins (51/56), (3) five proteins involved in ribosome biogenesis/maturation, (4) 10 factors modulating translation, (5) two enzymes introducing post-translational modifications in proteins, (6) seven factors facilitating protein folding and (7) 21 factors involved in protein export (signal peptidase, Lol system involved in lipoprotein export, general secretion Sec system, Tat translocation system and TAM translocation and assembly module specific for autotransporters) (Table 1). Furthermore, many other essential genes encode proteins involved in additional key processes such as (1) genome replication and maintenance

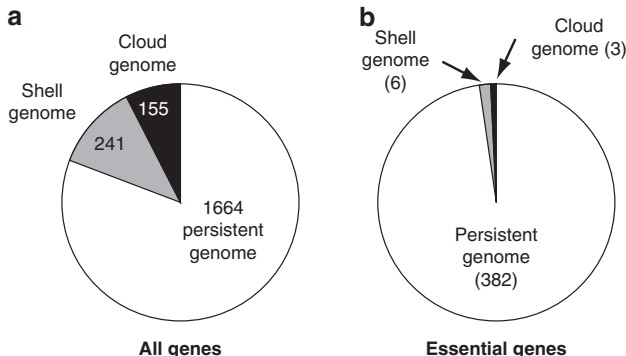

**Fig. 2 Partition of essential meningococcal genes into persistent, shell and cloud genomes.** To perform this analysis, we used the PPanGGOLiN[21] method as explained in the text. **a** Partition of the 2060 genes in the genome of *N. meningitidis* 8013: persistent (not highlighted), shell (highlighted in grey), and cloud (highlighted in black). The corresponding datasets are listed in Supplementary Data 5. **b** Partition of the subset of 391 essential genes identified in this study (same colour code than in **a**). The corresponding datasets are listed in Supplementary Data 5.

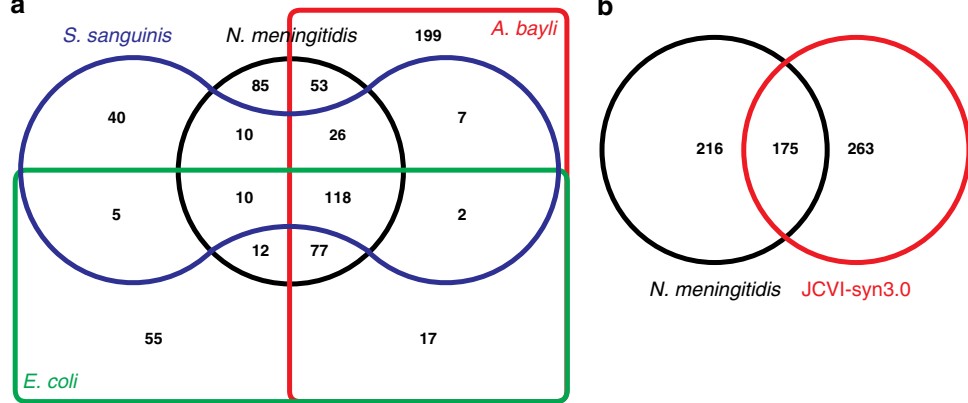

**Fig. 3 Comparison of *N. meningitidis* essential genome to that of other bacteria. a** Edwards–Venn diagram displaying overlaps between essential genes in bacteria in which complete libraries of mutants have been constructed: *N. meningitidis* (black line), *E. coli* (green line)[10], *A. baylyi* (red line)[12] and *S. sanguinis* (blue line)[13]. To perform this analysis, we queried the DEG database[65] of essential genes using our set of essential genes. The corresponding datasets are listed in Supplementary Data 6. **b** Comparison of *N. meningitidis* essential genome (black line) to JCVI-syn3.0 (red line), a synthetic *M. mycoides* designed with a minimal genome[22]. The corresponding datasets are listed in Supplementary Data 7.

**Table 1 Essential meningococcal genes listed by functional category/sub-category.**

| Functional category/sub-category | Number of genes | % of total |
|---|---|---|
| Gene/protein expression | 150 | 38.4 |
| Ribosomal protein | 51 | 13 |
| Aminoacyl-tRNA synthesis | 24 | 6.1 |
| Protein export | 21 | 5.4 |
| RNA modification/degradation | 19 | 4.9 |
| DNA transcription | 11 | 2.8 |
| Translation factors | 10 | 2.6 |
| Protein folding | 7 | 1.8 |
| Ribosome biogenesis/maturation | 5 | 1.3 |
| Protein post-translational modification | 2 | 0.5 |
| Genome/cell replication | 33 | 8.4 |
| Chromosome replication/maintenance | 20 | 5.1 |
| Cell division | 13 | 3.3 |
| Cell membrane/wall biogenesis | 54 | 13.8 |
| Peptidoglycan biosynthesis/recycling | 17 | 4.3 |
| LOS biosynthesis | 14 | 3.6 |
| Fatty acid biosynthesis | 12 | 3.1 |
| Phospholipids biosynthesis | 8 | 2 |
| UDP-GlcNAc biosynthesis | 3 | 0.8 |
| Cytosolic metabolism | 120 | 30.7 |
| Vitamins biosynthesis | 24 | 6.1 |
| Nucleotide biosynthesis | 15 | 3.8 |
| Aerobic respiration/cytochrome $c$ | 12 | 3.1 |
| Amino acid metabolism | 9 | 2.3 |
| Isoprenoid biosynthesis | 9 | 2.3 |
| Electron transport | 9 | 2.3 |
| Heme biosynthesis | 9 | 2.3 |
| Sugar metabolism | 9 | 2.3 |
| Fe-S cluster biosynthesis | 6 | 1.5 |
| CoA/ac-CoA biosynthesis | 6 | 1.5 |
| Ubiquinol biosynthesis | 4 | 1 |
| NAD/NADP biosynthesis | 3 | 0.8 |
| Lipoate biosynthesis | 2 | 0.5 |
| SAM biosynthesis | 1 | 0.3 |
| Phosphate metabolism | 1 | 0.3 |
| Sulfur metabolism | 1 | 0.3 |
| Unassigned | 34 | 8.7 |
| Total | 391 | 100 |

For this classification, we used predictions from MultiFun[24], MetaCyc[25], eggNOG[26], COG[27], FIGfam[28] and/or InterProScan[29]. The corresponding datasets are listed in Supplementary Data 8.

(20 proteins including subunits of the DNA polymerase, topoisomerases, gyrase, ligase etc.), (2) cell division (13 proteins), (3) peptidoglycan cell wall biogenesis (17 proteins) and (4) membrane biogenesis (37 proteins) (Table 1). The latter class includes all 12 proteins required for fatty acid biosynthesis, eight proteins required for the biosynthesis of phospholipids (phosphatidylethanolamine (PE) and phosphatidylglycerol (PG)), and 17 proteins involved in the biosynthesis and export of the lipo-oligosaccharide (LOS), which constitutes the external leaflet of the outer membrane. Finally, most of the remaining genes of known function (Table 1) are involved in energy generation, production of key metabolic intermediates (such as dihydroxyacetone phosphate (DHAP), or 5-phosphoribosyl diphosphate (PRPP)) and a variety of metabolic pathways leading to the biosynthesis of

(1) vitamins (B1, B2, B6 and B9), (2) nucleotides, (3) amino acids (notably *meso*-diaminopimelate (M-DAP) a component of the peptidoglycan), (4) isoprenoid, (5) heme, (6) CoA/ac-CoA, (7) NAD/NADP, (8) iron-sulfur (Fe-S) clusters, (9) ubiquinol, (10) lipoate and (11) *S*-adenosyl-methionine (SAM). Many of these compounds are cofactors and/or coenzymes known to be essential for the activity of many bacterial enzymes.

When essential genes were further integrated into networks and metabolic pathways—using primarily MetaCyc[25]—the above picture became dramatically simpler. As many as 91.3% of the meningococcal essential genes partitioned into just four major functional categories (Supplementary Fig. 2). Namely, (1) gene/protein expression, (2) genome/cell replication, (3) membrane/cell wall biogenesis and (4) cytosolic metabolism. Strikingly, for most multi-step enzymatic pathways, most, and often all, the corresponding genes were identified as essential (Fig. 4), although they are most often scattered throughout the genome. We view this as important quality control of our library. Only 34 essential genes (8.7%) could not be clearly assigned to these four categories. Critically, this functional partition is coherent with the one previously determined for the minimal synthetic bacterium JCVI-syn3.0[22]. As a result, we were able to generate a concise overview of the meningococcal essential genome in the context of the cell (Fig. 4), in which many of the above pathways are detailed and often linked by critical metabolic intermediates. This overview provides a useful blueprint for systems and synthetic biology and a global understanding of meningococcal biology.

**Essential genes that are part of the variable genome are conditionally essential.** A puzzling finding was that some essential genes, which could not be assigned to the above four functional groups, are part of shell/cloud genomes present at intermediate/low frequency in the meningococcus (Supplementary Data 5). Interestingly, many of the corresponding genes are located in regions of genome plasticity (RGP)—often thought to have been acquired by horizontal gene transfer—as confirmed by the recently described predictive method panRGP[30] that identifies RGP using pangenome data generated by PPanGGOLiN[21]. A panRGP analysis identified 32 RGP in the genome of strain 8013, encompassing a total of 348,885 bp (15.3% of the genome) (Supplementary Data 9). Interestingly, one of these genes offered a plausible explanation to the apparent paradox of essential genes that are not part of the persistent meningococcal genome. NMV_2289 is predicted to encode a DNA-methyltransferase, based on the presence of a DNA methylase N-4/N-6 domain (InterPro IPR002941). Since the neighbouring NMV_2288 is predicted to encode a restriction enzyme, the probable role of the NMV_2289 methyltransferase is to protect meningococcal DNA against degradation by this restriction enzyme. This would explain the lethal phenotype of the ΔNMV_2289 mutant. A closer examination of the other essential genes that are part of shell/cloud genomes and often found in RGP suggests that a similar scenario might be often applicable. We noticed that NMV_1478, which encodes a protein of unknown function in RGP_2, is likely to be co-transcribed with NMV_1479 that is predicted to encode the toxin of a toxin–antitoxin (TA) system[31] (Fig. 5a). Therefore, since a toxin and its cognate antitoxin are often encoded by closely linked genes, we hypothesised that NMV_1478 might be the antitoxin for the neighbouring NMV_1479 toxin. This would explain why NMV_1478 is essential, i.e. in its absence, the NMV_1479 toxin would kill the cell. If this is true, we reasoned that it should be possible to delete NMV_1478 together with NMV_1479, which was attempted. As predicted, in contrast to NMV_1478 which could not be deleted individually despite

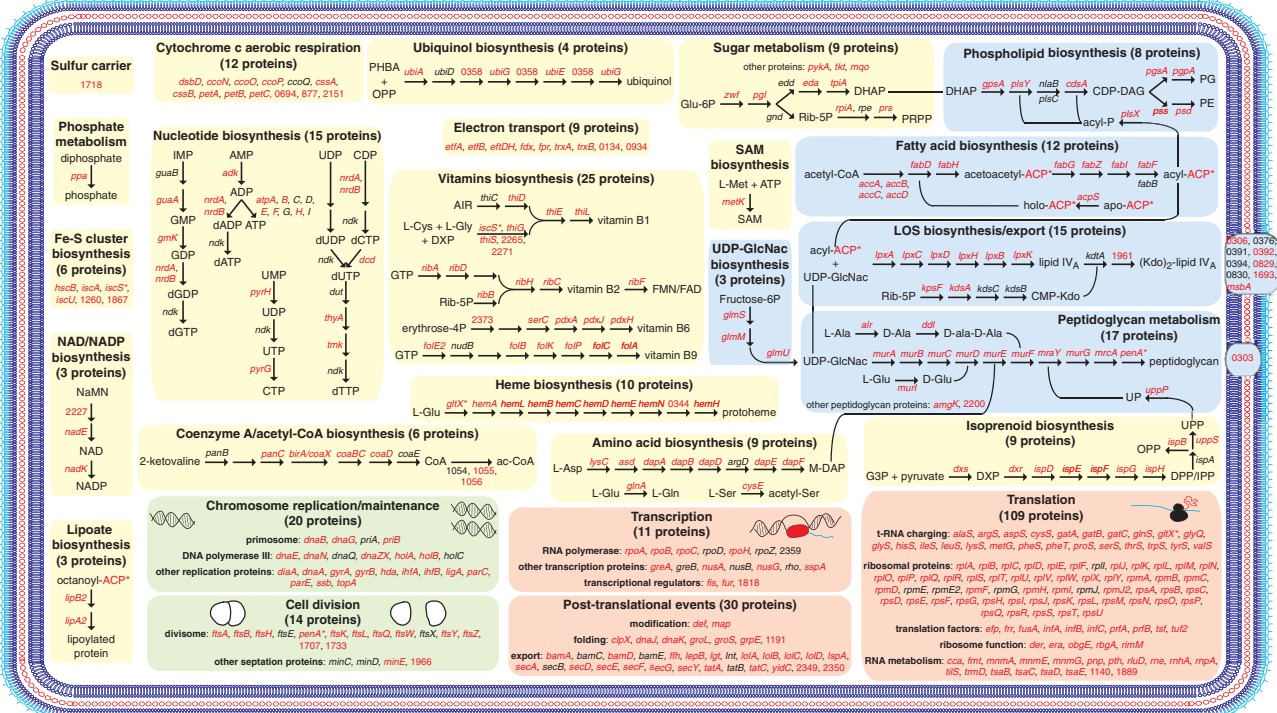

**Fig. 4 Concise cellular overview of the essential meningococcal genome.** Essential genes were integrated into networks and metabolic pathways using primarily MetaCyc[25]. The four basic functional groups are highlighted using the same colour code as in Supplementary Fig. 2, i.e. orange (gene/protein expression), green (genome/cell replication), blue (cell membrane/wall biogenesis) and yellow (cytosolic metabolism). The 34 essential genes that could not be clearly assigned to one of these four categories are not represented on the figure. Genes involved in the different reactions are indicated by their name or NMV_ label, in red when essential, in black when dispensable. *Genes involved in more than one pathway. Key compounds/proteins are abbreviated as follows. ACP acyl carrier protein, AIR aminoimidazole ribotide, CDP-DAG CDP-diacylglycerol, CMP-Kdo CMP-ketodeoxyoctonate, DHAP dihydroxyacetone phosphate, DPP dimethylallyl diphosphate, DXP 1-deoxyxylulose-5P, Fe-S iron-sulfur, FMN flavin mononucleotide, G3P glyceraldehyde-3P, Glu-6P glucose-6P, IMP inosine monophosphate, IPP isopentenyl diphosphate, LOS lipo-oligosaccharide, M-DAP *meso*-diaminopimelate, NaMN nicotinate ᴅ-ribonucleotide, OPP all-*trans*-octaprenyl diphosphate, PE phosphatidylethanolamine, PG phosphatidylglycerol, PHBA *p*-hydroxybenzoate, PRPP 5-phosphoribosyl diphosphate, Rib-5P ribulose-5P, SAM *S*-adenosyl-methionine, UDP-GlcNAc UDP-*N*-acetyl-glucosamine, UPP di-*trans*-poly-*cis*-undecaprenyl diphosphate. The corresponding datasets are listed in Supplementary Data 8.

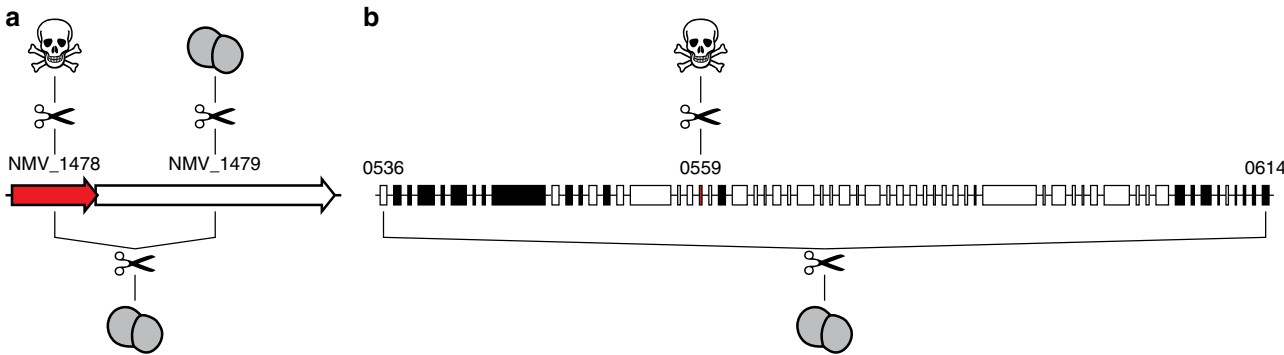

**Fig. 5 Essential genes in RGP—NMV_1479 and NMV_0559—are conditionally essential. a** Gene organisation of the putative antitoxin NMV_1478 (highlighted in red) with its neighbouring NMV_1479 toxin. Results of the mutagenesis (represented by scissors) are shown. Viable mutant (represented by a grey diplococcus); lethal phenotype (represented by a skull). **b** Gene organisation of the *tps* RGP (RGP_0) to which NMV_0559 (highlighted in red) belongs (Supplementary Data 9). Genes on the + strand are in white, genes on the—strand are in black. Results of the mutagenesis (scissors) are shown. Viable mutant (grey diplococcus); lethal phenotype (skull). In contrast to NMV_0559, each of the other target genes in the *tps* RGP could be mutated individually (not shown for readability).

repeated attempts, a double deletion mutant ΔNMV_1478/1479 could be readily obtained (Fig. 5a). This confirms that the NMV_1478 gene is essential only when the toxin product of NMV_1479 is also present. Such a scenario was also tested for NMV_0559, which is found in a large RGP of 86.8 kbp (RGP_0),

encompassing genes NMV_0527 to NMV_0618/0619 (Fig. 5b). We named this RGP *tps* because it contains multiple *tpsA* genes predicted to encode large haemagglutinin/haemolysin-related proteins of two-partner secretion systems[32]. Since NMV_0559 is the only essential gene in the *tps* RGP, we tested whether it might

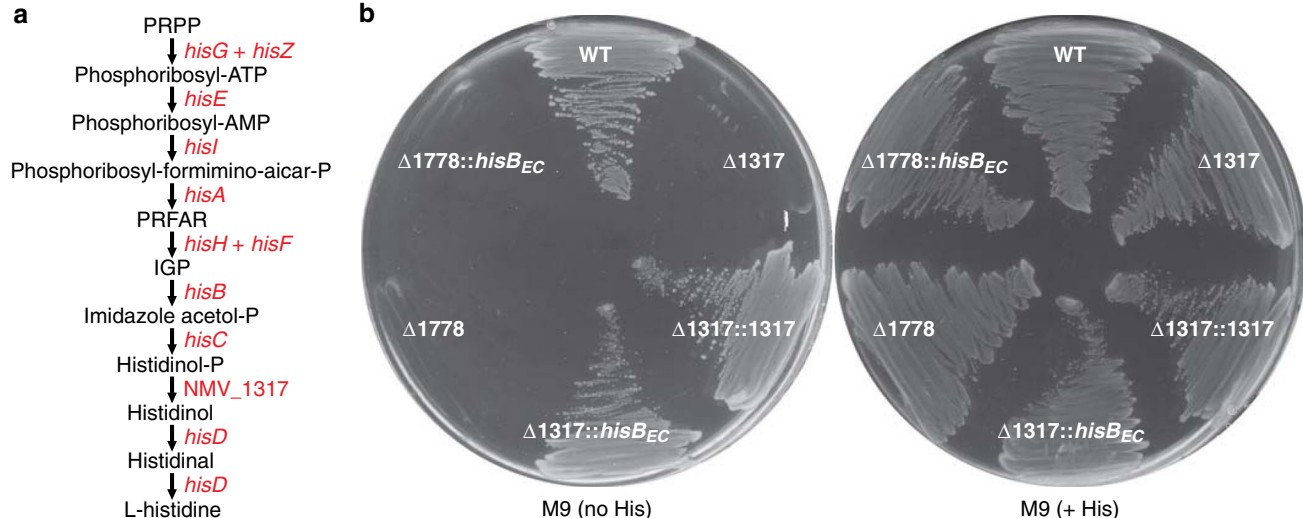

**Fig. 6 NMV_1317 encodes a novel histidinol-phosphatase. a** Histidine biosynthesis pathway in the meningococcus, with genes highlighted in red. PRPP 5-phosphoribosyl diphosphate, PRFAR phosphoribulosylformimino-AICAR-P, IGP erythro-imidazole-glycerol-P. **b** Growth on M9 minimal medium, with or without added histidine (His). The plates also contained 0.5 mM IPTG for inducing expression of the complementing genes. WT, strain 8013; Δ1317, ΔNMV_1317 mutant; Δ1317::1317, ΔNMV_1317 complemented with NMV_1317; Δ1317::$hisB_{EC}$, ΔNMV_1317 cross-complemented with $hisB_{EC}$ from *E. coli*, which encodes the unrelated histidinol-phosphatase present in this species; Δ1718, ΔNMV_1718 mutant; Δ1718::$hisB_{EC}$, control showing that ΔNMV_1718 (*hisH*) cannot be cross-complemented by $hisB_{EC}$. Source data are provided as a Source Data file.

be conditionally essential by attempting to delete most of RGP_0. As predicted, while NMV_0559 could not be deleted on its own despite repeated attempts, an 80 kbp Δ*tps* deletion in RGP_0 encompassing NMV_0559 could be readily obtained (Fig. 5b). This suggests that NMV_0559 is essential only in the presence of another gene in RGP_0, which remains to be identified. We predict that several other essential genes of unknown function that are not part of the persistent meningococcal genome may be similarly conditionally essential, including (1) NMV_1305 and NMV_1310 part of a putative prophage (RGP_1), (2) NMV_1541 the putative antitoxin for neighbouring NMV_1539 toxin, (3) NMV_1544 putative antitoxin for neighbouring NMV_1543 toxin, (4) NMV_1761 putative immunity protein against neighbouring MafB toxins (RGP_4), which represent a family of secreted toxins in pathogenic *Neisseria* species[33], (5) NMV_1918 part of a second *tps* RGP (RGP_7), (6) NMV_2010 the putative immunity protein against neighbouring bacteriocins (RGP_3) and (7) NMV_2333 and NMV_2335 putative immunity proteins in a second *maf* RGP (RGP_11).

**Filling holes in metabolic pathways: identification of a novel and widespread histidinol-phosphatase.** Another major utility of the NeMeSys 2.0 collection of mutants is to determine the function of genes of unknown function by reverse genetics. We, therefore, wished to illustrate this aspect. We noticed that the 10-step metabolic pathway leading to histidine biosynthesis presents an apparent "hole" (Fig. 6a). The "missing" enzyme in 8013 corresponds to histidinol-phosphatase (EC 3.1.3.15), which catalyses the dephosphorylation of histidinol-P into histidinol (Fig. 6a). The meningococcal genome does not have homologues of known histidinol-phosphatases[34], a feature it shares with many other bacterial species according to MetaCyc[25]. First, we excluded the possibility that the meningococcus might be auxotrophic for histidine by showing that 8013 can grow on M9 minimal medium without added histidine (Fig. 6b). Next, we sought to identify the unknown histidinol-phosphatase using NeMeSys 2.0, by specifically testing mutants in genes encoding putative phosphatases of unknown function, in search of a mutant that would not grow on M9 without added histidine. Using this approach, we identified

ΔNMV_1317 as a histidine auxotroph, growing on M9 plates only when histidine was added (Fig. 6b). The product of this gene was annotated as a putative hydrolase of unknown function, belonging to the IB subfamily of the haloacid dehalogenase superfamily of aspartate-nucleophile hydrolases (IPR006385). The corresponding sequences include a variety of phosphatases none of which is annotated as a histidinol-phosphatase. Complementation of ΔNMV_1317 mutant with NMV_1317 restored growth on M9 without added histidine (Fig. 6b), confirming that the auxotrophic phenotype was due to the mutation in NMV_1317. To confirm that NMV_1317 encodes the missing enzyme in histidine biosynthesis, we performed a cross-species complementation assay with the *hisB* gene from *E. coli* DH5α (*hisB_{EC}*), which encodes a histidinol-phosphatase unrelated to NMV_1317. Complementation of ΔNMV_1317 with *hisB_{EC}* restored growth on M9 without added histidine (Fig. 6b). Complementation was gene-specific, since *hisB_{EC}* could not complement the growth deficiency of a different histidine auxotrophic mutant in NMV_1778 (*hisH*) (Fig. 6b). Together, these results show that NMV_1317 defines a previously unrecognised class of histidinol-phosphatases, allowing us to fill a hole in the histidine biosynthesis pathway in the meningococcus. This finding, which illustrates the utility of NeMeSys 2.0 for annotating genes of unknown function by reverse genetics, has implications for other species lacking known histidinol-phosphatases. Indeed, NMV_1317 homologues are widespread in Betaproteobacteria and Gammaproteobacteria in which no histidinol-phosphatase has been identified according to MetaCyc[25]. We expect that NMV_1317 homologues will encode the elusive histidinol-phosphatase in a variety of Burkholderiales and Pseudomonadales, including many pathogenic species of *Bordetella*, *Pseudomonas* and *Burkholderia* (there is some evidence for this in *Burkholderia phytofirmans*[35]).

**Identification of a comprehensive set of genes involved in type IV pilus biology in *N. meningitidis*.** Another utility of NeMeSys 2.0 we wished to illustrate is the possibility to perform whole-genome phenotypic screens to identify all the genes responsible for a phenotype of interest. We chose to focus on type IV pili

**Table 2 Mutants in the complete NeMeSys 2.0 library of meningococcal mutants affected for two functions mediated by T4P: aggregation and twitching motility.**

| Mutated gene | Product | Aggregation | Twitching |
|---|---|---|---|
| Genes previously known to be involved in T4P biology in *N. meningitidis* | | | |
| *pilE* | Major pilin PilE | − | N/A |
| *pilT* | Type IV pilus retraction ATPase PilT | + (Irregular) | − |
| *pilF* | Type IV pilus extension ATPase PilF | − | N/A |
| *pilD* | Leader peptidase/*N*-methyltransferase PilD | − | N/A |
| *pilG* | Type IV pilus biogenesis protein PilG | − | N/A |
| *pilW* | Type IV pilus biogenesis lipoprotein PilW | − | N/A |
| *pilX* | Minor pilin PilX | − | N/A |
| *pilK* | Type IV pilus biogenesis protein PilK | − | N/A |
| *pilJ* | Type IV pilus biogenesis protein PilJ | − | N/A |
| *pilI* | Type IV pilus biogenesis protein PilI | − | N/A |
| *pilH* | Type IV pilus biogenesis protein PilH | − | N/A |
| *pilZ* | PilZ protein | − | N/A |
| *pilM* | Type IV pilus biogenesis protein PilM | − | N/A |
| *pilN* | Type IV pilus biogenesis protein PilN | − | N/A |
| *pilO* | Type IV pilus biogenesis protein PilO | − | N/A |
| *pilP* | Type IV pilus biogenesis lipoprotein PilP | − | N/A |
| *pilQ* | Type IV pilus secretin PilQ | − | N/A |
| Genes not previously known to be involved in T4P biology in *N. meningitidis* | | | |
| *tsaP* | Secretin-associated protein TsaP | − | N/A |
| NMV_1205 | Conserved hypothetical periplasmic protein | − | N/A |
| NMV_2228 | Putative carbonic anhydrase | +/− | N/A |

The other 1569 mutants can form aggregates and exhibit twitching motility. *N/A* not assayable because twitching motility could only be assessed in mutants forming aggregates. The corresponding datasets are listed in Supplementary Data 10.

(T4P), which are pivotal virulence factors in the meningo-coccus[36], for two reasons. First, T4P, are ubiquitous in prokaryotes, which has made them a hot topic for research for the past 40 years[37]. Second, although many aspects of T4P biology remain incompletely understood, most genes composing the multi-protein machinery involved in the assembly of these filaments and/or their multiple functions have been identified, including in the meningococcus that is one of the mainstream T4P models[36]. Hence, we could readily benchmark the results of our screen against previous mutational analyses, especially for the recovery rate of expected mutants. The two T4P-linked phenotypes we decided to study are the formation of bacterial aggregates and twitching motility. Both phenotypes can be simultaneously assessed—allowing thus a dual screen—by observing the mutants growing in liquid medium by phase-contrast microscopy[38]. We thus scored the 1589 mutants for the presence of round aggregates, and for the continuous and vigorous jerky movements of cells within these aggregates, which corresponds to twitching motility[38] (Supplementary Data 10). This analysis revealed that 20 mutants (1.2%) present phenotypic defects in these two T4P-linked phenotypes (Table 2). Significantly, we identified 100% of the expected single mutants in 17 *pil* genes known to affect these phenotypes in the meningococcus[38–40]. This is another quality control of the NeMeSys 2.0 collection of mutants, which confirms excellent correlation between phenotype and genotype, and shows that when a robust screening method is used all the genes involved in a given phenotype can be identified. Furthermore, we could readily identify mutants such as Δ*pilT* able to form (irregular) aggregates, but showing no twitching motility[38]. PilT is known to encode the motor powering pilus retraction, which is directly responsible for twitching motility[41]. No other mutant in the library exhibited a similar phenotype. Strikingly, we also identified mutants in three genes not previously associated with T4P biology in *N. meningitidis* (Table 2), which was unexpected considering that T4P has been studied for decades in this species. TsaP has been shown in the closely related pathogen *Neisseria gonorrhoeae* to interact with the secretin PilQ[42], which forms a

pore in the outer membrane through which T4P translocate onto the cell surface. TsaP has a poorly understood role in T4P biology in *N. gonorrhoeae* and *Myxococcus xanthus*[42], but has apparently no role in *Pseudomonas aeruginosa*[43]. In addition, we identified NMV_1205 and NMV_2228, which to the best of our knowledge have never been previously linked to T4P biology. NMV_1205 is predicted to encode a periplasmic protein of unknown function and is found mainly in Neisseriales, where it is widespread. NMV_2228 is predicted to encode a carbonic anhydrase, an enzyme catalysing the reversible hydration of carbon dioxide[44].

Next, we analysed the three new mutants in detail using an approach previously validated in the meningococcus with many *pil* genes[38,45]. We first tested whether piliation was affected in the mutants by purifying T4P using a procedure in which filaments sheared by vortexing are precipitated using ammonium sulfate[38]. Pilus preparations, obtained from equivalent numbers of cells, were separated by SDS-PAGE and either stained using Coomassie blue, or tested by immunoblotting using an anti-PilE antibody. This revealed the major pilin PilE as a 17 kDa species (Fig. 7a) and confirmed that all three mutants were piliated. However, when compared to the WT, pilus yields differed between the mutants. While piliation in ΔNMV_2228 was apparently normal, it was decreased in Δ*tsaP* and ΔNMV_1205. The decrease was dramatic in ΔNMV_1205, where filaments could be detected only by immunoblotting (Fig. 7a). Quantification of piliation using a whole-cell ELISA procedure[38] confirmed these findings (Fig. 7b) and showed that piliation levels were 40, 31 and 154% of WT, in Δ*tsaP*, ΔNMV_1205 and ΔNMV_2228, respectively. Therefore, none of the corresponding proteins is required for T4P biogenesis.

We then further characterised the new mutants for T4P-linked functions. Since it is known that in some meningococcal mutants piliation and/or aggregation defects can be restored when filament retraction is abolished by a concurrent mutation in *pilT*[38,45], we tested if that might be the case for the new mutants. First, we showed that aggregation, which is abolished in Δ*tsaP* and ΔNMV_1205 and dramatically affected in ΔNMV_2228,

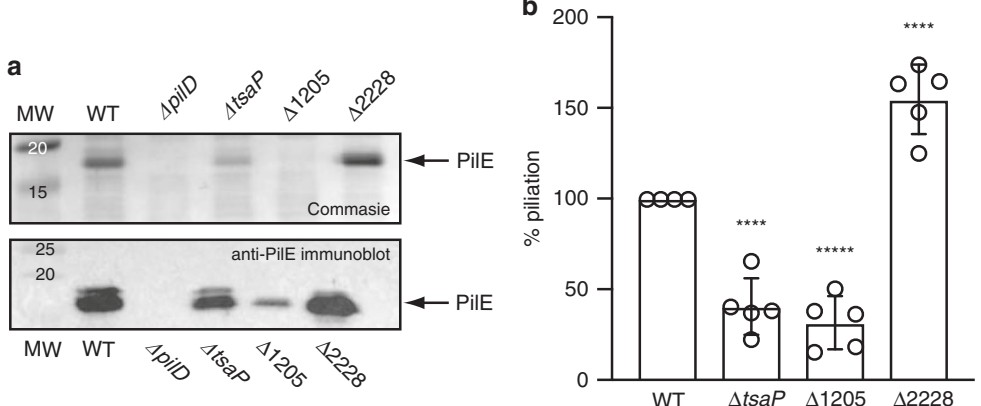

**Fig. 7 Assaying piliation in the mutants in genes not previously associated with T4P biology in *N. meningitidis*.** The WT strain and a non-piliated Δ*pilD* mutant were included as positive and negative controls, respectively. Δ*tsaP*, Δ*tsaP* mutant; Δ1205, ΔNMV_1205 mutant; Δ2228, ΔNMV_2228 mutant. **a** T4P purified using a shearing/precipitation method were separated by SDS-PAGE and either stained with Coomassie blue (upper panel) or analysed by immunoblotting using an antibody against the major pilin PilE (lower panel)[59]. Samples were prepared from equivalent numbers of cells and identical volumes were loaded in each lane. MW molecular weight marker lane, with values in kDa. Source data are provided as a Source Data file. **b** T4P were quantified by whole-cell ELISA using a monoclonal antibody specific for the filaments of strain 8013[58]. Equivalent numbers of cells, based on $OD_{600}$ readings, were applied to the wells of microtiter plates. Results are expressed in % piliation (ratio to WT) and are the average ± standard deviations from five independent experiments. Source data are provided as a Source Data file. For statistical analysis, one-way ANOVA followed by Dunnett's multiple comparison tests were performed (****$P < 0.0001$).

could be restored when mutants were complemented with the corresponding WT alleles (Fig. 8a). This confirmed that the phenotypic defects in these mutants were indeed due to the mutations in the above three genes. Similarly, some aggregation was restored in Δ*tsaP*/Δ*pilT* and ΔNMV_1205/Δ*pilT* (ΔNMV_2228/Δ*pilT* could not be constructed), which harboured a concurrent mutation in *pilT* that encodes the retraction motor PilT. However, aggregates were morphologically distinct from the round aggregates formed by the WT or the irregular aggregates formed by Δ*pilT*. These findings suggest that *tsaP* and NMV_1205 participate in the formation of aggregates by counterbalancing PilT-mediated pilus retraction. Lastly, we checked whether the mutants were affected for another T4P-linked phenotype, competence for DNA uptake, which makes the meningococcus naturally transformable. We quantified competence in the three mutants as described[46] by transformation to rifampicin resistance (Fig. 8b). We found that Δ*tsaP* was almost as transformable as the WT, ΔNMV_1205 showed a 26-fold decrease in transformation, while competence was completely abolished in ΔNMV_2228 (explaining why the ΔNMV_2228/Δ*pilT* double mutant could not be constructed). These findings show that *tsaP*, NMV_1205 and NMV_2228 contribute to the fine-tuning of multiple functions mediated by T4P in *N. meningitidis*.

Taken together, these findings are of significance in two ways. The identification of three new genes playing a role in T4P biology (how exactly remains to be understood), will contribute to a better understanding of these filaments. This has broad implications because T4P and T4P-related filamentous nanomachines are ubiquitous in Bacteria and Archaea[47]. In addition, the identification of new genes contributing to phenotypes that have been extensively studied in several species for the past 30 years is unambiguous evidence of the potential of NeMeSys 2.0 to lead to a global phenotypic profiling of the meningococcal genome.

**Concluding remarks.** Here, we describe the construction of a complete collection of defined mutants in *N. meningitidis*, which has been fully integrated in our modular NeMeSys toolbox[16], accessible online in MicroScope[23]. Furthermore, we illustrate

NeMeSys 2.0 utility for tackling the challenge of genes of unknown function, following a variety of approaches. Although they were limited at this stage to a few different biological properties, these experiments provide significant information on meningococcal biology, with possible implications for many other species. They also clearly demonstrate the potential of NeMeSys 2.0 to make a significant contribution to ongoing efforts directed towards a comprehensive understanding of a bacterial cell. Such potential is further amplified by several useful properties of *N. meningitidis* such as (1) its small genome with limited functional redundancy, which in species with larger genomes can obscure the link between phenotype and genotype, (2) its hardy nature with a robust metabolism allowing it to grow on minimal medium, (3) its taxonomy making it the first Betaproteobacteria in which a complete library of mutants is available and (4) the fact that it is a major human pathogen, which allows addressing virulence properties absent in non-pathogenic model species. This latter point makes the meningococcus a prime model for the identification of new means for controlling bacterial pathogens, which would have practical implications for human health.

The other achievement in the present study is the identification of the essential meningococcal genome, comprising 391 genes that could not be disrupted. The finding that more than 90% of these genes are associated with just four basic biological functions, helped us generate a coherent concise cellular overview of the meningococcal essential metabolism. We surmise that most, if not all, of the remaining essential genes of unknown function, which are conserved in other bacterial species, will be involved in either expression of genome information, preservation of genome information, cell membrane structure/function, or cytosolic metabolism (in particular the production of key metabolic intermediates, vitamins and crucial cofactors/coenzymes). Our cellular overview provides a useful starting point for systems biology, which together with previous genome-scale metabolic network models[48,49], could help us progress towards a global understanding of meningococcal biology. This, in turn, would have widespread implications, because many of the meningococcal genes, especially the essential ones, are widely conserved.

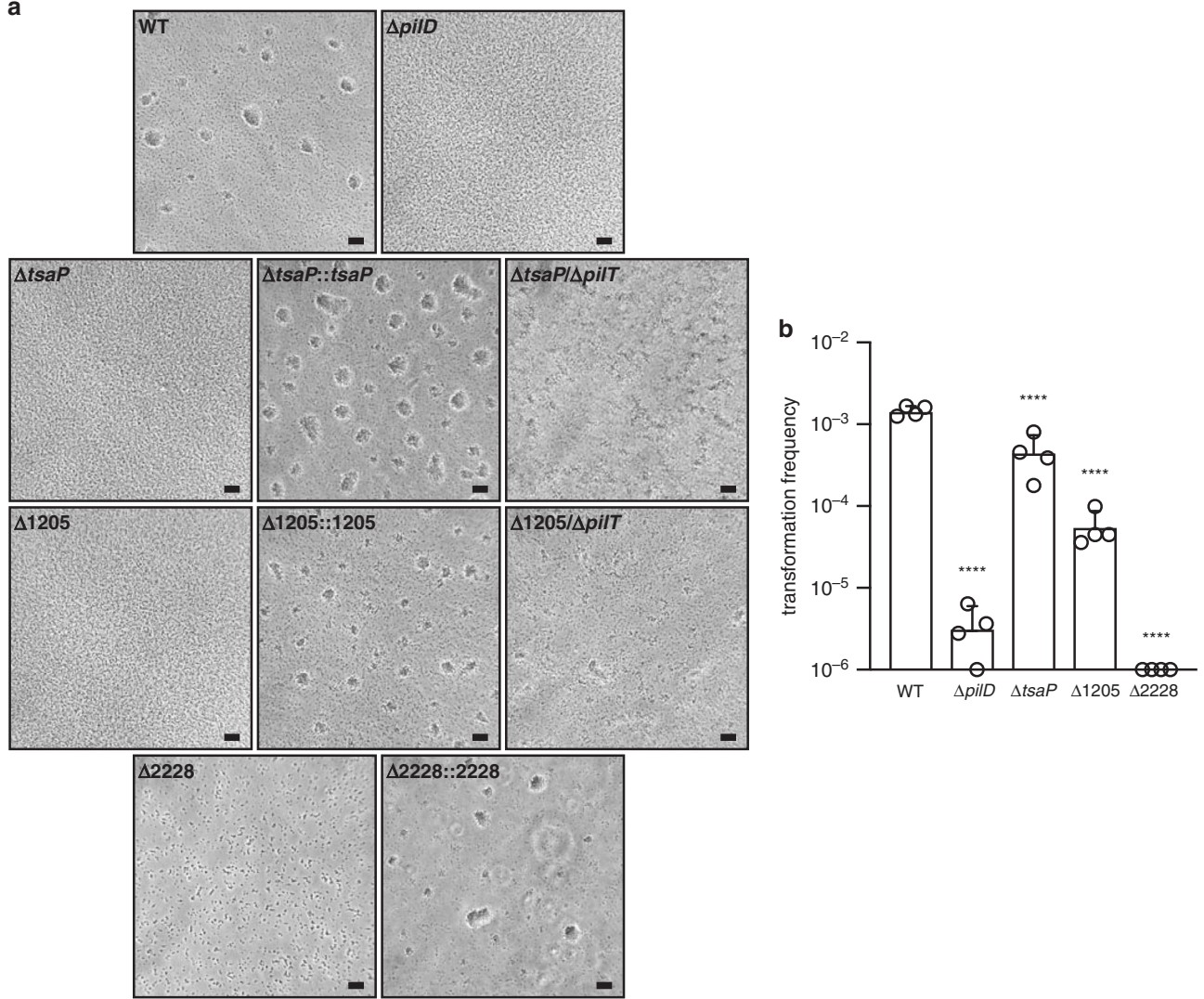

**Fig. 8 Functional analysis of the mutants in genes not previously associated with T4P biology in *N. meningitidis*.** The WT strain and a non-piliated *pilD* mutant were included as positive and negative controls, respectively. **a** Aggregation in liquid culture as assessed by phase-contrast microscopy[38]. Δ*tsaP*:: *tsaP*, Δ*tsaP* complemented with *tsaP*; Δ*tsaP*/Δ*pilT*, double mutant in *tsaP* and *pilT*; Δ1025::1205, ΔNMV_1205 complemented with NMV_1205; Δ1205/ Δ*pilT*, double mutant in NMV_1205 and *pilT*; Δ2228::2228, ΔNMV_2228 complemented with NMV_2228. Scale bar, 20 μm. Source data are provided as a Source Data file. **b** Quantification of the competence for DNA transformation. Equivalent numbers of recipient cells were transformed using a *rpoB* PCR product containing a point mutation leading to rifampicin resistance. Results are expressed as transformation frequencies and are the average ± standard deviations from four independent experiments. Source data are provided as a Source Data file. For statistical analysis, one-way ANOVA followed by Dunnett's multiple comparison tests were performed (****$P < 0.0001$).

## Methods

**Strains and growth conditions**. All the *N. meningitidis* strains that were generated and used in this study were derived from a highly adhesive variant—sometimes called clone 12 or 2C43—of the clinical isolate 8013[50]. This serogroup C strain, which belongs to the sequence type 177 and clonal complex ST-18, has been previously sequenced[16]. Meningococci were routinely grown on plates with GC agar Base (GCB) (Difco) containing 5 g/l agar (all chemicals were from Sigma unless stated otherwise) and Kellogg's supplements (4 g/l glucose, 0.59 μM thiamine hydrochloride, 12.37 μM Fe(NO₃)₃·9H₂O, 68.4 μM ʟ-glutamine). Plates were incubated overnight (O/N) at 37 °C in a moist atmosphere containing 5% $CO_2$. Alternatively, we used agar plates with M9 minimal medium (4 g/l glucose, 4.78 mM $Na_2HPO_4$, 2.2 mM $KH_2PO_4$, 1.87 mM $NH_4Cl$, 8.56 mM NaCl, 2 mM $MgSO_4$, 0.1 mM $CaCl_2$). When required, plates contained 100 μg/ml kanamycin, 3 μg/ml erythromycin, 5 μg/ml rifampicin, 25 μg/ml ʟ-histidine and/or 0.5 mM isopropyl-β-ᴅ-thiogalactopyranoside (IPTG) (Merck). Strains were stored at −80 °C in 10% glycerol in liquid GC (15 g/l protease peptone No. 3, 23 mM $K_2HPO_4$, 7.34 mM $KH_2PO_4$, 85.6 mM NaCl). *E. coli* TOP10 and DH5α were grown at 37 °C in liquid or solid lysogeny broth (LB), which contained 100 μg/ml spectinomycin or 50 μg/ml kanamycin, when appropriate.

**Construction of strains**. Genomic DNA from *N. meningitidis* and *E. coli* strains were prepared using the Wizard Genomic DNA Purification kit (Promega) following the manufacturer's instructions. Plasmid DNA from *E. coli* strains was purified using the QIAprep Spin Miniprep Kit (Qiagen) following the manufacturer's instructions. Bacteria were transformed as follows. *N. meningitidis* is naturally competent for transformation. A loopful of bacteria grown on GCB plates was resuspended in liquid GC containing 5 mM $MgCl_2$ (GC transfo), 200 μl was aliquoted in the well of a 24-well plate, and DNA was added. After incubation for 30 min at 37 °C on an orbital shaker, 0.8 ml GC transfo was added to the wells and the plates were further incubated for 3 h at 37 °C, without shaking. Transformants were selected on GCB plates containing suitable antibiotics. For the transformation of *E. coli*, ultra-competent cells were prepared as described elsewhere[51] and transformed by a standard heat shock procedure[52]. Transformants were selected on LB plates containing the suitable antibiotic.

The construction of the NeMeSys 2.0 library of meningococcal mutants followed a two-step procedure explained in the Results section (Supplementary Fig. 1). First, we selected a subset of potentially suitable mutants from an archived library of Tn mutants with sequence-defined Tn insertion sites, which was described previously[16,17]. The corresponding genomic DNAs were PCR-verified

using primers flanking the Tn insertion sites (Supplementary Data 2), designed using Primer3[53] either in batch (version 1.1.4) or manually (version 4.1.0). When the mutants were confirmed to be correct, the corresponding mutations were re-transformed in 8013, the mutants were PCR-verified once again and stored at −80 °C. The remaining target genes, for which no Tn mutants were available, were systematically submitted to non-polar targeted mutagenesis using a validated no-cloning method[20]. In brief, using high-fidelity PfuUltra II Fusion HS DNA Polymerase (Agilent) and two sets of specific primers (F1/R1 and F2/R2), we amplified 500-750 bp PCR products upstream and downstream from each target gene, respectively. The R1 and F2 primers were consecutive, non-overlapping and chosen within the target gene (excluding the first and last 30%) (Supplementary Fig. 1). These primers contained 20-mer overhangs complementary to the F3/R3 primers used to amplify the 1518 bp kanamycin resistance cassette present in the Tn (Supplementary Data 2), which contains a DNA uptake sequence necessary for efficient transformation in the meningococcus[54] (Supplementary Fig. 1). In the first step, three PCR products were amplified separately using F1/R1, F2/R2 and F3/R3 pairs of primers. The first two PCRs contained a 4-fold excess of outer primers F1 and R2[55]. The three products were then combined (3.3 μl of each) and spliced together by PCR using the excess F1/R2 primers added in the first reaction[55]. The sPCR products were directly transformed into *N. meningitidis*. For each successful transformation, two colonies were isolated and verified by PCR using F1/R2. When transformations yielded no transformants, they were repeated at least three times with different sPCR products. This method was also used to construct several polymutants in which we deleted the 80 kbp RGP_0, the TA system NMV_1478/1479, and three repeated prophages (NMV_1286/1294, NMV_1387/1398, and NMV_1412/1418). For each gene that was successfully mutated, one mutant was individually stored at −80 °C in glycerol, and the tubes were ordered according the gene NMV_ label. Likewise, corresponding genomic DNAs that allow easy re-transformation of these mutations (even in other *Neisseria* strains), were individually ordered in Eppendorf tubes and stored at −20 °C. Several mutants, as well as the WT strain, were verified by WGS. Sequencing was performed by MicrobesNG on an Illumina sequencer, using standard Nextera protocols. An average of 122 Mb of read sequences (between 80 and 179 Mb) were obtained for each sequencing project, representing an average 53-fold genome coverage. The corresponding Illumina reads have been deposited in the European Nucleotide Archive.

The Δ*tsaP*/Δ*pilT* and ΔNMV_1205/Δ*pilT* double mutants were constructed by transforming Δ*tsaP* and ΔNMV_1205 with genomic DNA extracted from a *pilT* mutant disrupted by a cloned *ermAM* cassette conferring resistance to erythromycin[56]. ΔNMV_2228/Δ*pilT* could not be constructed because both mutations abolish competence. For complementation assays, complementing genes were amplified using specific indF/indR primers (Supplementary Data 2), with overhangs introducing flanking *Pac*I sites for cloning, and a ribosome binding site in front of the gene. The corresponding PCR products were cloned into pCR8/GW/TOPO (Invitrogen), verified by Sanger sequencing and subcloned into pGCC4[57]. This placed the genes under the transcriptional control of an IPTG-inducible promoter, within an intragenic region of the gonococcal chromosome conserved in *N. meningitidis*[57]. The resulting plasmids were transformed in the desired meningococcal mutant, leading to ectopical insertion of the complementing gene. We thus constructed ΔNMV_1317::NMV_1317, ΔNMV_1317::*hisB*$_{EC}$, ΔNMV_1718::*hisB*$_{EC}$, Δ*tsaP*::*tsaP*, ΔNMV_1205::NMV_1205 and ΔNMV_2228::NMV_2228. Since ΔNMV_2228 is not competent, the last strain was constructed by first transforming the corresponding pGCC4 derivative into the WT strain, before transforming the ΔNMV_2228 mutation. Expression of the complementing genes in *N. meningitidis* was induced by growing the strains on GCB plates containing 0.5 mM IPTG.

**Dual screen for mutants affected for aggregation and/or twitching motility**. *N. meningitidis* mutants grown O/N on GCB plates were resuspended individually with a 200 μl pipette tip in the wells of 24-well plates containing 500 μl pre-warmed RPMI 1640 with L-glutamine (PAA Laboratories), supplemented with 10% heat-inactivated fetal bovine serum Gold (PAA Laboratories). Plates were incubated for ~2 h at 37 °C. Aggregates forming on the bottom of the wells were visualised by phase-contrast microscopy using a Nikon TS100F microscope[38]. Digital images of aggregates were recorded using a Sony HDR-CX11 camcorder mounted onto the microscope. In parallel, we scored twitching motility by observing whether bacteria in aggregates exhibited continuous and vigorous jerky movement[38]. This movement is abolished in a Δ*pilT* mutant, in which the T4P retraction motor that powers twitching motility is not produced anymore.

**Detection and quantification of T4P**. Pilus purification by ammonium sulfate precipitation was carried out as follows. Bacteria were grown O/N on GCB plates were first resuspended in 1.5 ml ethanolamine buffer (150 mM ethanolamine, 65 mM NaCl) at pH 10.5. Filaments were sheared by vortexing at maximum speed for 1 min before OD$_{600}$ was adjusted to 9–12 using ethanolamine buffer (in 1.5 ml final volume). Bacteria were then pelleted by centrifugation at 17,000 × *g* at 4 °C during 10 min. The supernatant (1.35 ml) was recovered, topped to 1.5 ml with ethanolamine buffer. This step was repeated once, and filaments were precipitated for ~1 h at room temperature (RT) by adding 150 μl ethanolamine buffer saturated with ammonium sulphate. Filaments were then pelleted by centrifugation at 17,000×*g* at

4 °C during 15 min. Pellets were rinsed once with 100 μl Tris-buffered saline at pH 8 and finally resuspended in 100 μl of Laemmli buffer (BioRad) containing β-mercaptoethanol.

Piliation was quantified by performing whole-cell ELISA[38] using the 20D9 mouse monoclonal antibody that is specific for the T4P in strain 8013[58]. Bacteria grown O/N on GCB plates were resuspended in PBS, adjusted to OD$_{600}$ 0.1 and heat-killed during 1 h at 56 °C. Serial 2-fold dilutions were then aliquoted (100 μl) in the wells of 96-well plates. Each well was made in triplicate. The plates were dried O/N at RT in a running safety cabinet. The next day, wells were washed seven times with washing solution (0.1% Tween 80 in PBS), before adding 100 μl/well of 20D9 antibody (diluted 1/1000 in washing solution containing 5% skimmed milk). Plates were incubated 1 h at RT and then washed seven times with washing solution. Next, we added 100 μl/well of Amersham ECL anti-mouse IgG HRP-linked whole antibody (GE Healthcare) diluted 1/10,000. Plates were incubated 1 h at RT and then washed seven times with washing solution. We then added 100 μl/well of TMB solution (Thermo Scientific) and incubated the plates for 20 min at RT in the dark. Finally, we stopped the reaction by adding 100 μl/well of 0.18 M sulfuric acid, before reading the plates at 450 nm using a plate reader. Statistical analyses were performed with Prism (GraphPad Software). Comparisons were done by one-way ANOVA, followed by Dunnett's multiple comparison tests. An adjusted *P* value < 0.05 was considered significant (*$P < 0.05$, **$P < 0.01$, ***$P < 0.001$, ****$P < 0.0001$).

**SDS-PAGE, Coomassie staining and immunoblotting**. Purified T4P were separated by SDS-PAGE using 15% polyacrylamide gels. Gels were stained using Bio-Safe Coomassie stain (BioRad) or blotted to Amersham Hybond ECL membranes (GE Healthcare) using standard molecular biology techniques[52]. Blocking, incubation with primary/secondary antibodies and detection using Amersham ECL Plus reagents (GE Healthcare) were done following the manufacturer's instructions. The primary antibody was a previously described rabbit anti-PilE serum (1/2500)[59], while the secondary antibody (1/10,000) was a commercial Amersham ECL anti-rabbit IgG HRP-linked whole antibody (GE Healthcare). Full scan Coomassie-stained gels and/or immunoblots can be seen in the Source Data file.

**Quantifying transformation in the meningococcus**. Since the tested mutants are Km$^R$, we tested their competence by transforming them to Rif$^R$ using DNA from a mutant of 8013 spontaneously resistant to rifampicin[60]. We first amplified by PCR, using rpoF/rpoR primers, a 1172 bp internal portion of *rpoB*, which usually contains point mutations[61] leading to Rif$^R$. This PCR product was cloned in pCR8/GW/TOPO and sequenced, which revealed a single point mutation, leading to a His → Tyr substitution at position 553 in RpoB. This pCR8/GW/TOPO derivative was used to amplify the PCR product used for quantifying competence, which was done as follows. Bacteria grown O/N on GCB plates were resuspended in pre-warmed liquid GC transfo at an OD$_{600}$ of 0.1. The number of bacterial cells in this suspension was quantified by performing colony-forming unit (CFU) counts. We mixed 200 μl of bacterial suspension with 100 ng of transforming DNA in the wells of a 24-well plate. After incubating for 30 min at 37 °C on an orbital shaker, 0.8 ml GC transfo was added to the wells and the plates were further incubated during 3 h at 37 °C without shaking. Transformants were selected by plating appropriate dilutions on plates containing rifampicin and by counting, the next day, the number of Rif$^R$ CFU. Transformation frequencies are expressed as % of transformed recipient cells. Statistical analyses were performed with Prism (GraphPad Software). As above, comparisons were done by one-way ANOVA, followed by Dunnett's multiple comparison tests.

**Bioinformatics**. The Qiagen CLC Genomics Workbench software was used for WGS analysis. In brief, Illumina reads from each strain were mapped onto the published sequence of 8013[16], and good quality and high frequency (>70%) base changes and small deletions/insertions were identified. Large deletions/insertions were identified using annotated assemblies (N50 between 38,959 and 45,097 bp), which were visualised in Artemis (version17.0.1)[62], by performing pairwise sequence alignments using DNA Strider (version 3.5 z1)[63].

The genome of strain 8013 and a total of 108 complete genomes from the RefSeq database[64] (last accessed June 8th, 2020) (Supplementary Data 5) was used to compute the *N. meningitidis* pangenome using the PPanGGOLiN software[21] (version 1.1.85). The original annotations of the genomes have been kept in order to compute gene families. In brief, PPanGGOLiN uses a statistical model to infer the pangenome classes (persistent, shell and cloud) based on both the presence/absence of gene families and genomic neighbourhood information[21]. The options "–use pseudo" and "–defrag" were used to consider pseudogenes in the gene family computation, and to associate fragmented genes with their original gene family, respectively. The other parameters have been used with default values to compute the persistent, shell and cloud partitions. The PPanGGOLiN results were then used to predict regions of genomic plasticity using the panRGP module within PPanGGOLiN (version 1.1.85), with default options[30].

Datasets and annotations generated during this study have been stored within MicroScope[23]. This publicly accessible web interface can be used to visualise genomes (simultaneously with synteny maps in other microbial genomes), perform

queries (by BLAST or keyword searches) and download datasets/annotations in a variety of formats (including EMBL and GenBank).

**Reporting summary**. Further information on research design is available in the Nature Research Reporting Summary linked to this article.

## Data availability

Whole-genome sequencing data that support the findings in this study have been deposited in the European Nucleotide Archive under accession number PRJEB39197. All the datasets generated during this study are either included in this paper and its Supplementary Information files, or available in MicroScope (http://mage.genoscope.cns.fr/microscope/mage/viewer.php?O_id=99). The RefSeq database (http://www.ncbi.nlm.nih.gov/refseq/) was used to access annotations of publicly available genomes of *N. meningitidis*. Lists of essential genes in other bacteria were obtained from the DEG repository (http://tubic.tju.edu.cn/deg/). Source data are provided with this paper.

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

## Acknowledgements

This work was supported by funding from the Wellcome Trust (092290/Z/10/Z) to V.P. The France Génomique and French Bioinformatics Institute national infrastructures— funded as part of Investissement d'Avenir programme managed by the Agence Nationale pour la Recherche (contracts ANR-10-INBS-09 and ANR-11-INBS-0013)—are acknowledged for support of the MicroScope annotation platform. We thank Angelika Gründling (Imperial College London) and Christoph Tang (University of Oxford) for critical reading of the manuscript.

## Author contributions

V.P. was responsible for conception and supervision of the work, interpretation of data and writing of the manuscript. A.M., I.G., A.C. and V.P. performed the experimental studies. A.B. and D.V. performed computational studies.

## Competing interests

The authors declare no competing interests.
