## [Peer Review File · Nature Communications]

REVIEWER COMMENTS

Reviewer #1 (Remarks to the Author):

Summary

The manuscript by Muir et al reports the construction of a complete set of gene knockouts in *Neisseria meningitidis* strain 8013 using two different mutagenesis schemes. The first scheme uses individual transposon insertions picked from the previously developed partial mutant library (NeMeSys) each transposon was used that mapped to roughly the middle of each gene in this collection. The second approach used in vitro construction of insertional mutants of the remainder of predicted genes and DNA transformation to introduce nonessential mutants into the library. Analysis of both viable and nonviable (ie, essential) mutants provides new information about metabolism, active restriction/modification genes, addiction modules, other toxins, and type 4 pilus assembly. This report provides an important tool for the study of this organism and the analysis of the mutants reveals several important biological processes and new genes involved in those processes. The text is clearly written, and the figures are mostly clearly presented. Both the tool and the new information provided will have high impact on the study of this bacterium and related organisms, but there are several issues that should be clarified.

Specific Issues

1. Since the partial NeMeSys collection of transposon mutants was published and this new library of mutants is more complete and uses a different methodology for over half the mutants, it seems like a new name should be used. Perhaps NeMeSys2?
2. A map/cartoon of the two types of insertions and their features would be helpful for the reader.
3. It is never discussed in the text whether all of the 'essential gene' constructs had a transformation uptake sequence, or if there was a DUS in the construct. If not, there is a chance that some are not essential under these conditions.
4. Nowhere in the manuscript are operons dealt with. Does either the transposon or the antibiotic resistance gene interrupt or alter the expression of genes in the same operon as the target gene? In fact neither of the terms 'operon' or 'polarity' are found in the text and should be addressed.
5. Page 6, line 119: The results report a new annotation and the definition of many small RNAs. It does not seem that the small RNAs were mutated and this fact should be explicitly stated.
6. Page 7, line 167: It might be more accurate to report the 391 genes are essential under these conditions.
7. Page 8, line 191: What exactly defines the cutoffs for shell and cloud genes is not explained. Is there a strain percentage cutoff for each class? Moreover, could the authors comment on why pIC1 is in the shell and pIC2 is in the cloud? Is it clear that they and other gene families (opa, pilE/S) are assembled and annotated properly in the genomic sequences?
8. Page 13, lines 313-14: It would be good (but not essential) to know what gene in the 80 kb region is required to allow deletion of NMV_0559.
9. Figure 3: Part A of this figure is difficult to process. Perhaps use pairwise comparisons as was done in part B..
10. Figure 6: Are any of these genes (particularly 0559) in an operon?
11. Figure 9: The text reports that the delta1205/pilT strain restores aggregation, but the micrograph only shows a partial restoration. This partial effect should be reported and discussed.
12. Tinsley 2004 and Sinha 2008 reported that DsbA is necessary for pilus biogenesis. If a dsbA mutant was not revealed in the pilus function screens, a discussion of this difference should be added to the discussion.
13. The text uses a number of words that could be removed since they are unnecessary and distract from the message. The ones I noticed are: important, highly (several times), breathtaking, bonanza, very more, far more, workhorse, highly (several times).

Hank Seifert

Reviewer #2 (Remarks to the Author):

Muir and colleagues describe a complete collection of archived, sequence-defined mutant strains in the pathogen *Neisseria meningitidis*. This work extends their previous effort (also called NeMySys), where they generated transposon insertion mutants in ~1,000 different genes. To extend the mutant collection, they systematically deleted the genes that were not hit in the older work. The authors then describe the essential gene set of *N. meningitidis* and its properties (conservation in other bacteria, functional category enrichment), and more importantly assay the collection for conditional phenotypes among nonessential genes. Interestingly, the authors were able to fill a previously unknown gap in histidine biosynthesis and also identified some new genes involved in type IV pili biology. Overall, the work will be interest to the *N. meningitidis* community, in particular groups interested in comprehensive genetic screens or targeted analysis of single genes.

Major comments:

* The authors frequently describe their data, and in particular the essential gene set, as being synonymous with the “minimal genome”. While the description of the essential gene set (under the conditions the authors used to select the mutants) is an important advance for *N. meningitidis* (and of relevance to the genes that comprise a theoretical minimal *N. meningitidis* genome), the reader is left with the impression that the authors view these 391 genes as the minimal genome. There is no evidence for this. To experimentally describe a minimal *N. meningitidis* genome, one would need to start removing multiple genes, chunks of the genome, etc. Or synthesize a genome from the ground up. Obviously, both of these are big efforts BUT groups are using both of these approaches to address this challenge. And one of the early lessons from these studies includes instances where two non-essential genes cannot both be removed from the cell (maybe they’re functionally redundant paralogs, or biology is just quite complex and cannot be accurately predicted). So, the authors should not conflate these terms (essential genes and minimal genome). I recommend restricting the mention of minimal genome to a brief mention in the Discussion, in the context that the elucidation of the essential gene set in the current study is of relevance if one wanted to design and construct a minimal *N. meningitidis* genome (although I’m not sure why one would want to do this in this particular system, maybe because the genome is on the smaller side and the microbe appears pretty easy to genetically engineer).

* The authors should explicitly state in the Introduction and Results section (at least) that the mutant collection is a combination of transposon insertion mutants and gene deletion mutants. It took me a while to pick up on this.

Specific comments:

line 19 - Not sure what “theoretical considerations” means in the context of this paper.

line 24 - This study doesn’t describe a “view of minimal genome of this species”. See above.

line 29 “all previously known” genes involved in type IV pili? This is too absolute (see below).

line 32 - “These findings have widespread implications in bacteria” is a bit of an oversell. The genetic resources generated are valuable and the hisB/pili results are interesting, but these are a limited number of new insights into gene function.

line 45 - I recommend updating ref #2 to the E. coli "Y-ome paper" recently published in NAR...(<https://pubmed.ncbi.nlm.nih.gov/30698741/>)

line 66- The primary downsides of Tn-seq are that follow-up studies on single genes are often difficult (especially if good tools for targeted genome engineering are not developed), some assays are hard to perform in a pooled assay format (secreted factors can be exchanged among the mutant strains in trans, additionally there are instances where one (or a few) mutation dominates the library and obscures the majority of the strains).

line 83-84 - Again, I don't think you can call the set of essential genes the "minimal genome".

line 103: The previous collection was called also "NeMeSys", which is a bit confusing. The current work is an expansion of the original NeMeSys and now includes many targeted gene deletion strains. Is another name suitable, to distinguish from the previous effort? NeMeSys2.0?

line 108-109: Should be an analysis of the essential gene complement, not the "minimal genome".

lines 115-116: State more explicitly that a systematic re-annotation of the N. meningitidis 8013 genome was done as part of this study. Relatedly, is the updated genome sequence in a public database?

Change "Supplemental spreadsheets" to Supplemental Tables (or the like to match journal specifications)? Also, the file names of the Supplemental spreadsheets were not evident in the excel files.

line 140-142: What is the rationale behind retransforming the old Tn mutants to make new strains? Why didn't the authors just use these old strains to start the new collection?

line 153: In general, what fraction of the target gene was deleted? The phrasing here is ambiguous. In most instances, was the entire open reading frame deleted?

line 181: change "minimal genome" to "essential genome".

lines 248-254: Are any of these compounds expected to be in the rich growth media used to select the mutants? If so, then it's likely that 8013 is incapable of transporting these compounds.

line 255: Delete "Remarkably". Essential genes in bacteria are highly conserved and these metacyc functional categories are ridiculously broad (i.e. "cytosolic metabolism"). These results are not surprising at all.

line 272-3: Most of the examples given in this section are not about conditional essentiality, which would imply that the gene is indispensable in another growth condition, but rather that the genes are not essential in a different genetic background (i.e. when another gene is deleted). I suggest rephrasing this.

line 327: "determining" -> "determine".

line 361-364: Based on the data from:
<https://pubmed.ncbi.nlm.nih.gov/29769716/>

A homolog from Burkholderia phytofirmans PsJN (BPHYT_RS03625) is also an auxotroph that is rescued by histidine. This would be good to mention, as it provides additional support for the authors' finding.

Data accessed from:

http://fit.genomics.lbl.gov/cgi-bin/domains.cgi?orgId=BFirm&locusId=BPHYT_RS03625

line 366-7: "complete set" is too extreme (and doesn't leave any wiggle room, for example if two functionally redundant genes are later found to be involved in type IV pilus biology). Recommend changing "complete" to something like "comprehensive". For example, how quantitative are the twitching motility and aggregate phenotypic assays? Are these the judgement call of someone looking under the microscope? Also note that in the follow-up studies, two of the three new hits were not completely required for pili formation, while the 3rd was not required at all.

line 474-475. This study did not identify the minimal meningococcal genome, just the essential gene set (see above). Do the authors believe that a cell with just these 391 genes would be viable? There's no evidence of this.

line 477: Again, delete "remarkably".

line 483 (and earlier): I'm not sure what "cellular panorama" means in the context of this study.

lines 486-488: This is an oversell. Essential genes are highly conserved across all free-living bacteria (this isn't a new finding), and there are many, many papers describing essential and conditionally essential gene sets in bacteria. Why is this paper special and what unique insight does it offer on the "theoretical implications for our understanding of the bases of cellular life..."

Line 855: In Table 1, indicate where these functional categories are derived from.

Line 861: In Table 2, for Twitching motility assays, exactly what does not assayable mean? My understanding is that these mutants were twitching motility minus.

Figure 1, change "targets" to "targeted" (two times, leftmost pie chart).

Line 890. Change "minimal genome" to "essential gene set".

Figure 4, in the legend describe the origin of these functional categories.

Line 923: As described above, I recommend changing "conditionally essential" to "genetic background dependent" or something like that.

Figure 8A, The genotype labels are off-centered for the top gel.

- Adam Deutschbauer (LBNL and UC Berkeley)

Reviewer #1

The manuscript by Muir et al reports the construction of a complete set of gene knockouts in Neisseria meningitidis strain 8013 using two different mutagenesis schemes. The first scheme uses individual transposon insertions picked from the previously developed partial mutant library (NeMeSys) each transposon was used that mapped to roughly the middle of each gene in this collection. The second approach used in vitro construction of insertional mutants of the remainder of predicted genes and DNA transformation to introduce nonessential mutants into the library. Analysis of both viable and nonviable (i.e. essential) mutants provides new information about metabolism, active restriction/modification genes, addiction modules, other toxins, and type 4 pilus assembly. This report provides an important tool for the study of this organism and the analysis of the mutants reveals several important biological processes and new genes involved in those processes. The text is clearly written, and the figures are mostly clearly presented. Both the tool and the new information provided will have high impact on the study of this bacterium and related organisms, but there are several issues that should be clarified.

We thank the Reviewer for his very positive assessment of our work.

1. Since the partial NeMeSys collection of transposon mutants was published and this new library of mutants is more complete and uses a different methodology for over half the mutants, it seems like a new name should be used. Perhaps NeMeSys2?

This is a valid point, which has also been made by Reviewer #2 (see point 10). We have therefore renamed, as suggested, our complete collection of mutants NeMeSys 2.0 throughout the manuscript.

2. A map/cartoon of the two types of insertions and their features would be helpful for the reader.

This has been done. We have added a Supplementary Fig. 1 describing the two types of mutations we have used.

3. It is never discussed in the text whether all of the 'essential gene' constructs had a transformation uptake sequence, or if there was a DUS in the construct. If not, there is a chance that some are not essential under these conditions.

The cassette used for mutagenesis indeed contains a DNA uptake sequence (Pelicic *et al.* 2000. *J. Bacteriol.*), which is necessary for efficient transformation in the meningococcus. This important point has now been spelled out in the text.

4. Nowhere in the manuscript are operons dealt with. Does either the transposon or the antibiotic resistance gene interrupt or alter the expression of genes in the same operon as the target gene? In fact, neither of the terms 'operon' or 'polarity' are found in the text and should be addressed.

We have extensive experimental evidence from this (see answer to point 10) and previous studies (see Georgiadou *et al.* 2012. *Mol. Microbiol.*) that mutations constructed using the above cassette are non-polar. This has now been mentioned in the text.

5. Page 6, line 119: The results report a new annotation and the definition of many small RNAs. It does not seem that the small RNAs were mutated and this fact should be explicitly stated.

It was mentioned in the manuscript that we targeted only protein-coding genes, hence non-coding RNAs were not included. We have now made this point clear right from the Abstract.

6. Page 7, line 167: It might be more accurate to report the 391 genes are essential under these conditions.

This has been done.

7. Page 8, line 191: What exactly defines the cut-offs for shell and cloud genes is not explained. Is there a strain percentage cut-off for each class? Moreover, could the authors comment on why piC1 is in the shell and piC2 is in the cloud? Is it clear that they and other gene families (opa, pilE/S) are assembled and annotated properly in the genomic sequences?

There is no predefined cut-off to determine the above classes. The PPanGGOLiN method that we used is based on a statistical model to infer the above classes within a pangenome, according to both the presence/absence of a gene families, and information on their genomic

neighbourhood. This has been clarified in the Materials and methods section. As mentioned in the manuscript, while the annotation of 8013 has been extensively curated manually, we used the original annotations of other genomes in RefSeq to compute gene families, which means that incorrect annotations in RefSeq could have an impact. This could explain rare instances of unexpected gene partitions between persistent, shell and cloud genomes. Actually, for a limited number of genes, the PPanGGOLiN partitioning was changed when the analysis was performed with many more meningococcal genomes. However, most of these genomes are unfinished (*i.e.* some genes were missing or were fragmented in a large portion of these genomes), which is why we restricted our analysis to complete genomes. Moreover, the 80 % aa identity threshold used in PPanGGOLiN to define homologous families is not well adapted for more variable genes such as *pilC* (many PilC hits display between 60% to 80% aa identity). Paralogous genes, such as *pilC1* and *pilC2*, are also known to be a confounding factor, which is why PPanGGOLiN results should be considered with caution for those. However, these observations do not have a significant impact on our general conclusions reported in the manuscript, which were based on the PPanGGOLiN analysis.

8. Page 13, lines 313-314: It would be good (but not essential) to know what gene in the 80 kb region is required to allow deletion of NMV_0559.

Although interesting, this would be extremely difficult to achieve because multiple repeats in the 80 kbp *tps* RGP (actually most of the RGP is formed of repeats) make it a very difficult target for mutagenesis. It is only after tedious efforts, and countless inconclusive attempts, that we managed to make individual mutants in all the corresponding genes in this RGP. This is the reason we decided to delete the whole *tps* region to demonstrate that NMV_0559 is conditionally essential.

9. Figure 3: Part A of this figure is difficult to process. Perhaps use pairwise comparisons as was done in part B.

Venn diagrams showing pairwise comparisons would lead to a significant loss of information. The Edwards-Venn display mode that we used is the only one that works reasonably well for multiple comparisons. We have therefore tried to make this figure easier to process by extensively modifying its legend.

10. Figure 6: Are any of these genes (particularly NMV_0559) in an operon?

We guess this is a point about possible polar effects like point 4 above. NMV_0559 is likely to be in an operon with the upstream gene (NMV_0558) since they are only separated by 4 bp. There is no gene immediately downstream NMV_0559, hence no possible polar effect to explain the phenotype of the mutant. As for NMV_1478, it is clearly in an operon with NMV_1479 since the two genes overlap by 4bp and are co-transcribed (Heidrich *et al.* 2017. Nucleic Acids Res). The phenotype of the NMV_1478 mutant (antitoxin) is another evidence that the mutations we have constructed are non-polar. If there was polarity Δ NMV_1478 mutant would behave as a double mutant Δ NMV_1478/1479 and would be viable, which is not what we observed.

11. Figure 9: The text reports that the Δ NMV_1205/ Δ *pilT* strain restores aggregation, but the micrograph only shows a partial restoration. This partial effect should be reported and discussed.

This has been done.

12. Tinsley 2004 and Sinha 2008 reported that *DsbA* is necessary for pilus biogenesis. If a *dsbA* mutant was not revealed in the pilus function screens, a discussion of this difference should be added to the discussion.

We agree that the term "complete" to describe our T4P screen was a tad too strong, which was also pointed by Reviewer #2 (see points 5 and 22). This term has therefore been changed to "comprehensive". We did not expect to identify *dsbA* for the same reason we did not identify mutants in *pilC*, because there are paralogs of these genes in the meningococcus genome. It would therefore be necessary to make polymutants to see a phenotype.

13. The text uses a number of words that could be removed since they are unnecessary and distract from the message. The ones I noticed are: important, highly (several times), breathtaking, bonanza, very more, far more, workhorse, highly (several times).

These changes have been done.

Reviewer #2

*Muir and colleagues describe a complete collection of archived, sequence-defined mutant strains in the pathogen *Neisseria meningitidis*. This work extends their previous effort (also called NeMeSys), where they generated transposon insertion mutants in ~1,000 different genes. To extend the mutant collection, they systematically deleted the genes that were not hit in the older work. The authors then describe the essential gene set of *N. meningitidis* and its properties (conservation in other bacteria, functional category enrichment), and more importantly assay the collection for conditional phenotypes among nonessential genes. Interestingly, the authors were able to fill a previously unknown gap in histidine biosynthesis and also identified some new genes involved in type IV pili biology. Overall, the work will be of interest to the *N. meningitidis* community, in particular groups interested in comprehensive genetic screens or targeted analysis of single genes.*

We thank the Reviewer for his very positive assessment of our work.

*1. The authors frequently describe their data, and in particular the essential gene set, as being synonymous with the "minimal genome". While the description of the essential gene set (under the conditions the authors used to select the mutants) is an important advance for *N. meningitidis* (and of relevance to the genes that comprise a theoretical minimal *N. meningitidis* genome), the reader is left with the impression that the authors view these 391 genes as the minimal genome. There is no evidence for this. To experimentally describe a minimal *N. meningitidis* genome, one would need to start removing multiple genes, chunks of the genome, etc. Or synthesize a genome from the ground up. Obviously, both of these are big efforts BUT groups are using both of these approaches to address this challenge. And one of the early lessons from these studies includes instances where two non-essential genes cannot both be removed from the cell (maybe they're functionally redundant paralogs, or biology is just quite complex and cannot be accurately predicted). So, the authors should not conflate these terms (essential genes and minimal genome). I recommend restricting the mention of minimal genome to a brief mention in the Discussion, in the context that the elucidation of the essential gene set in the current study is of relevance if one wanted to design and construct a minimal *N. meningitidis* genome (although I'm not sure why one would want to do this in this particular system, maybe because the genome is on the smaller side and the microbe appears pretty easy to genetically engineer).*

We agree that this is an important semantic point, and that the term "minimal genome" was not always used appropriately in our manuscript. Although essential genes defined by systematic mutagenesis indeed constitute the bulk of a minimal genome, the paper by Hutchison *et al.* (2016. *Science*) makes it clear by using whole-genome design and complete chemical synthesis that these genes fail to produce a viable cell. To obtain a viable *Mycoplasma* bacterium with a minimal genome, it was necessary to retain quasi-essential genes as well (leading to severe growth impairment when mutated), and to avoid deleting pairs of redundant genes for essential functions. We have therefore clarified this point in the manuscript.

2 The authors should explicitly state in the Introduction and Results section (at least) that the mutant collection is a combination of transposon insertion mutants and gene deletion mutants. It took me a while to pick up on this.

This has now been clearly spelled out early in the manuscript to avoid confusing the reader. We have also added a Supplementary Fig. 1 describing the different types of mutations we have generated.

3. line 19: Not sure what "theoretical considerations" means in the context of this paper.

"Theoretical" means fundamental, as opposed to practical. We have therefore changed this to "fundamental" throughout the text.

4. line 24: This study doesn't describe a "view of minimal genome of this species". See above.

This has been corrected.

5. line 29: "all previously known" genes involved in type IV pili? This is too absolute (see below).

This has been corrected.

6. line 32: *“These findings have widespread implications in bacteria” is a bit of an oversell. The genetic resources generated are valuable and the hisB/pili results are interesting, but these are a limited number of new insights into gene function.*

This has been corrected.

7. line 45: *I recommend updating ref #2 to the E. coli “Y-ome paper” recently published in NAR (<https://pubmed.ncbi.nlm.nih.gov/30698741/>).*

This has been done.

8. line 66: *The primary downsides of Tn-Seq are that follow-up studies on single genes are often difficult (especially if good tools for targeted genome engineering are not developed), some assays are hard to perform in a pooled assay format (secreted factors can be exchanged among the mutant strains in trans, additionally there are instances where one (or a few) mutation dominates the library and obscures the majority of the strains).*

This has been corrected.

9. line 83-84: *Again, I don’t think you can call the set of essential genes the “minimal genome”.*

This has been corrected.

10. line 103: *The previous collection was called also “NeMeSys”, which is a bit confusing. The current work is an expansion of the original NeMeSys and now includes many targeted gene deletion strains. Is another name suitable, to distinguish from the previous effort? NeMeSys2.0?*

This is a valid point, which has also been made by Reviewer #1 (see point 1). We have therefore renamed our complete collection of mutants NeMeSys 2.0 throughout the manuscript, as suggested.

11. line 108-109: *Should be an analysis of the essential gene complement, not the “minimal genome”.*

This has been corrected.

12. lines 115-116: *State more explicitly that a systematic re-annotation of the N. meningitidis 8013 genome was done as part of this study. Relatedly, is the updated genome sequence in a public database?*

The sentence has been changed as suggested. As mentioned in the text, the sequencing data have been deposited in the European Nucleotide Archive (PRJEB39197), while the re-annotation can be downloaded from the publicly accessible MicroScope platform (http://mage.genoscope.cns.fr/microscope/mage/viewer.php?O_id=99).

13. *Change “Supplemental spreadsheets” to Supplemental Tables (or the like to match journal specifications)? Also, the file names of the Supplemental spreadsheets were not evident in the excel files.*

This has been done.

14. line 140-142: *What is the rationale behind retransforming the old Tn mutants to make new strains? Why didn’t the authors just use these old strains to start the new collection?*

As shown for other ordered libraries of Tn mutants in which the Tn insertion sites have been sequence-defined by high-throughput experiments, a significant number of assignments are found to be incorrect when checked (Held *et al.* 2012. J Bacteriol). In order to construct a complete NeMeSys 2.0 collection of mutants of high quality, it was therefore crucial to verify that assignments in the original NeMeSys library of Tn mutants were accurate.

15. line 153: *In general, what fraction of the target gene was deleted? The phrasing here is ambiguous. In most instances, was the entire open reading frame deleted?*

As mentioned in the Materials and methods the internal primers for the deletion mutagenesis (R1 and F2 primers) were designed using Primer3 as consecutive, non-overlapping and within the target gene (excluding the first and last 30 %). This means that in none of the

mutants was the entire ORF deleted and that the fraction of the target gene that was deleted was highly variable (from a few %, up to 40% of the length of the gene) depending on the primers that were designed. This was decided, among other considerations, because the base composition of the 5' and 3' regions of meningococcal genes makes them very poor target for primer design. This point has been clarified and we have added a Supplementary Fig. 1 to describe the two types of mutations composing NeMeSys 2.0 (see point 2 by Reviewer #1).

16. line 181: change "minimal genome" to "essential genome".

This has been done.

17. lines 248-254: Are any of these compounds expected to be in the rich growth media used to select the mutants? If so, then it's likely that 8013 is incapable of transporting these compounds.

The GCB medium used to grow the meningococcus is an undefined medium with complex ingredients such as proteose peptone and corn starch, which makes it impossible to speculate if it contains some of the compounds we have identified as essential.

18. line 255: Delete "Remarkably". Essential genes in bacteria are highly conserved and these MetaCyc functional categories are ridiculously broad (i.e. "cytosolic metabolism"). These results are not surprising at all.

"Remarkably" has been deleted. Although potentially broad, the cytosolic metabolism functional category is composed of genes involved only in a limited number of pathways. What is remarkable is that for most multi-step enzymatic pathways often all the corresponding genes were identified as essential, and we could generate a coherent concise cellular overview of the meningococcal essential metabolism.

19. line 272-3: Most of the examples given in this section are not about conditional essentiality, which would imply that the gene is indispensable in another growth condition, but rather that the genes are not essential in a different genetic background (i.e. when another gene is deleted). I suggest rephrasing this.

Conditionally essential genes are genes essential only under certain conditions. Although as mentioned by the Reviewer, this most often applies to growth conditions, it can also apply to the role of genetic background (see Martínez-Carranza *et al.* 2018. FEMS Microbiology Letters). We have therefore clarified this point in the text.

20. line 327: "determining" > "determine".

This has been done.

21. line 361-364: Based on the data from <https://pubmed.ncbi.nlm.nih.gov/29769716>, a homolog from *Burkholderia phytofirmans* PsJN (BPHYT_RS03625) is also an auxotroph that is rescued by histidine. This would be good to mention, as it provides additional support for the authors' finding. Data accessed from http://fit.genomics.lbl.gov/cgi-bin/domains.cgi?orgId=BFirm&locusId=BPHYT_RS03625.

This has been mentioned.

22. line 366-7: "complete set" is too extreme (and doesn't leave any wiggle room, for example if two functionally redundant genes are later found to be involved in type IV pilus biology). Recommend changing "complete" to something like "comprehensive". For example, how quantitative are the twitching motility and aggregate phenotypic assays? Are these the judgement call of someone looking under the microscope? Also note that in the follow-up studies, two of the three new hits were not completely required for pili formation, while the 3rd was not required at all.

The term "complete" has been corrected to "comprehensive", which was also suggested by Reviewer #1 (see point 12). It is important to mention here that the phenotypic assays we have used, which we have long-standing experience with, although qualitative give unambiguous, yes or no answers. As for the last part of the comment, it is known that mutants can affect T4P-mediated properties without affecting piliation (we have published many papers on this topic), hence the phenotype of the mutants in the three new genes is not unheard of.

23. line 474-475: *This study did not identify the minimal meningococcal genome, just the essential gene set (see above). Do the authors believe that a cell with just these 391 genes would be viable? There's no evidence of this.*

This has been corrected. Please see answer to point 1.

24. line 477: *Again, delete "remarkably".*

This has been done.

25. line 483 (and earlier): *I'm not sure what "cellular panorama" means in the context of this study.*

"Cellular panorama" was meant as concise overview within the context of the cell. This has been clarified.

26. lines 486-488: *This is an oversell. Essential genes are highly conserved across all free-living bacteria (this isn't a new finding), and there are many, many papers describing essential and conditionally essential gene sets in bacteria. Why is this paper special and what unique insight does it offer on the "theoretical implications for our understanding of the bases of cellular life..."*

We have rephrased this sentence.

27. Line 855: *In Table 1, indicate where these functional categories are derived from.*

It was mentioned in the text that genes were classified in specific functional categories using predictions from MultiFun, MetaCyc, eggNOG, COG, FIGfam, and/or InterProScan. This is now mentioned in the legend to Table 1 as well.

28. Line 861: *In Table 2, for twitching motility assays, exactly what does not assayable mean? My understanding is that these mutants were twitching motility minus.*

The assay we used can only assess twitching motility in mutants capable of forming aggregates, hence the N/A qualifier for mutants in Table 1 that do not form aggregates. However, the Reviewer is right, twitching motility is known to be abolished in a majority of these mutants that do not produce T4P.

29. Figure 1: *change "targets" to "targeted" (two times, leftmost pie chart).*

This has been done.

30. Line 890: *Change "minimal genome" to "essential gene set".*

This has been done.

31. Figure 4: *in the legend describe the origin of these functional categories.*

It was mentioned in the text that essential genes were further integrated into networks and metabolic pathways using primarily MetaCyc. This is now mentioned in the legend to Fig. 4 as well.

32. Line 923: *As described above, I recommend changing "conditionally essential" to "genetic background dependent" or something like that.*

Please see answer to point 19.

33. Figure 8A: *The genotype labels are off-centred for the top gel.*

This has been corrected.